# R-Cut: Enhancing Explainability in Vision Transformers with Relationship Weighted Out and Cut

**DOI:** 10.3390/s24092695

**Published:** 2024-04-24

**Authors:** Yingjie Niu, Ming Ding, Maoning Ge, Robin Karlsson, Yuxiao Zhang, Alexander Carballo, Kazuya Takeda

**Affiliations:** 1Graduate School of Informatics, Nagoya University, Nagoya 464-8603, Japankarlsson.robin@g.sp.m.is.nagoya-u.ac.jp (R.K.); alexander@g.sp.m.is.nagoya-u.ac.jp (A.C.); takeda@g.sp.m.is.nagoya-u.ac.jp (K.T.); 2Graduate School of Engineering, Gifu University, Gifu 501-1112, Japan; 3Tier IV Inc., Tokyo 140-0001, Japan

**Keywords:** visual explanation, vision transformer, post hoc explanation, class-specific explanation

## Abstract

Transformer-based models have gained popularity in the field of natural language processing (NLP) and are extensively utilized in computer vision tasks and multi-modal models such as GPT4. This paper presents a novel method to enhance the explainability of transformer-based image classification models. Our method aims to improve trust in classification results and empower users to gain a deeper understanding of the model for downstream tasks by providing visualizations of class-specific maps. We introduce two modules: the “Relationship Weighted Out” and the “Cut” modules. The “Relationship Weighted Out” module focuses on extracting class-specific information from intermediate layers, enabling us to highlight relevant features. Additionally, the “Cut” module performs fine-grained feature decomposition, taking into account factors such as position, texture, and color. By integrating these modules, we generate dense class-specific visual explainability maps. We validate our method with extensive qualitative and quantitative experiments on the ImageNet dataset. Furthermore, we conduct a large number of experiments on the LRN dataset, which is specifically designed for automatic driving danger alerts, to evaluate the explainability of our method in scenarios with complex backgrounds. The results demonstrate a significant improvement over previous methods. Moreover, we conduct ablation experiments to validate the effectiveness of each module. Through these experiments, we are able to confirm the respective contributions of each module, thus solidifying the overall effectiveness of our proposed approach.

## 1. Introduction

Explainable machine learning has garnered significant attention in recent years. It refers to the ability of a machine learning model to provide an easily understandable causal relationship that explains the process of model prediction, thereby enhancing human confidence and facilitating model debugging for downstream tasks [1,2].

Explainability in deep learning models can be categorized into two main types [2]. The first category is intrinsic interpretability, which includes models with relatively simple structures like decision trees [3], logistic regression [4], and linear regression [5]. These models have transparent internal logic structures that can be readily understood during the model design process. However, their accuracy is generally lower compared to mainstream deep learning models. The second category is post hoc explainability, which involves employing various techniques to extract learned information from trained black box models, thereby enhancing their explainability. This type of explainability is particularly relevant for models with complex structures, such as convolutional neural networks (CNNs) [6,7,8,9,10] and vision transformers (ViTs) [11,12,13,14,15,16,17]. These models typically consist of billions of parameters, making it difficult to discern the direct causal relationships between the outputs and the internal structure of the model.

In the field of computer vision, a large amount of work has focused on increasing the explainability of CNNs by post hoc visualization of discriminative regions associated with targets in input images.

The emergence of vision transformers (ViTs) has revolutionized computer vision. Transformer-based methods such as Swin-transformer [15] and PVT [14] have surpassed traditional techniques and have achieved state-of-the-art (SOTA) performance in various computer vision tasks, including image classification, object detection, and semantic segmentation. Moreover, transformers have played a critical role in advancing multi-modal models such as CLIP [18], ALBEF [19], BLIP [20], and GLIP [21]. Additionally, transformers have been instrumental in the development of large language models (LLMs) [22], which have gained widespread popularity. However, as the application of transformers expands, the need for explainability methods becomes crucial. These methods enhance users’ confidence in model results and facilitate the debugging process, ultimately leading to improved performance in downstream tasks. Exploring explainability methods for transformers is a promising avenue to refine and optimize the performance of these models.

Despite these advancements, there are few contributions exploring the explainability of the ViT series of models. Most existing approaches only consider the direct use of the raw-attention map corresponding to the class token in the multi-head self-attention (MHSA) module to directly generate explainability maps in ViT [23,24,25]. However, these methods often adopt a class-agnostic approach, and the generated explainability maps tend to emphasize salient features while containing substantial noise. To address the noise problem associated with explainability methods based on the self-attention map, Abnar et al. proposed a method called attention rollout [26]. Although this approach improves the noise problem of raw attention to some extent, it often struggles to distinguish between true foreground and background regions.

Another approach was proposed by Chefer et al., it utilizes the deep Taylor decomposition principle to assign relevance and improve the problem mentioned above [27]. By combining the information from back-propagation gradients, this method achieves class-specific explainability. However, the presence of activation functions in the back-propagation process can lead to gradient vanishing and other issues, resulting in sparse and noisy explainability feature maps as outputs.

In our research, we propose a post hoc visualization explainability method called relationship weighted out and cut (R-Cut) with the objective of generating dense, low-noise, and class-specific explainability images for visual domain transformers and their derivative models. R-Cut consists of a two-stage extraction method, as illustrated in Figure 1. In the first stage, we propose a module called “Relationship Weighted Out (R-Out)” to extract the class-specific semantic features from the intermediate vectors. In the second stage, we propose a feature decomposition technique called “Cut” to decompose the class-specific semantic features into fine-grained foreground and background components.

To validate the effectiveness of our method, we conducted qualitative and quantitative experiments on the widely used ImageNet1K dataset [28] and compared the results with those of other SOTA methods. We also conducted experiments on the LRN dataset [29] designed for automated driving hazard alerts, which we created to test the explainability of our method in the presence of complex backgrounds. Furthermore, we performed ablation experiments to verify the effectiveness of the different modules proposed in our approach. Moreover, we conducted comparative experiments on various hyperparameters to validate their effectiveness. These comprehensive experiments aimed to provide evidence supporting the superiority of our method compared to existing approaches in terms of performance on standard benchmarks and its ability to handle complex scenarios.

This paper makes three main contributions:We propose a dense, low-noise, class-specific post hoc visualization explainability method for transformer-based models and their derivative models.We conducted various explainability tests on the largest image classification dataset in the world, demonstrating the superiority of our approach.We conducted extensive explainability experiments to validate the effectiveness of the proposed method in the context of autonomous driving scenarios with complex backgrounds. This contribution highlights the practical application of the method in real-world scenarios and demonstrates its ability to provide meaningful explanations even in challenging and intricate environments.

## 2. Related Work

### 2.1. CNN Explainability

In the field of computer vision, specifically for CNNs, a significant amount of research has focused on improving the interpretability of neural network models by generating post hoc visualizations of discriminative regions related to targets in input images [30,31,32,33,34,35,36,37]. There are three main groups of post hoc visualization methods that aim to enhance the explainability of neural network models in computer vision: CAM-based approaches, gradient-based approaches, and perturbation-based methods.

CAM-based approaches generate visual interpretation maps by linearly weighting the combination of activation maps from the last convolutional layer [30,31,33,34]. These approaches often have specific requirements for the network structure, such as the presence of a global pooling layer after the convolutional layer.

Gradient-based approaches [31,33,35,37] identify regions in input images that contribute most to the network’s output by back-propagating the gradient of the target category to the input image. However, this approach can suffer from gradient saturation and gradient vanishing issues due to the activation function, leading to noise in the generated gradient map. Additionally, Wang et al. [38] have demonstrated that the gradient-map-based approach can be susceptible to a false-confidence issue.

Perturbation-based approaches [39,40,41,42] determine the discriminative regions associated with the target by perturbing the input image and observing the change in confidence in the corresponding prediction. This approach provides more intuitive and easily understandable explainability maps. However, these methods often require the manual design of perturbation maps.

### 2.2. ViT Explainability

Currently, there remain few studies focusing on the explainability of methods belonging to the ViT family. Some approaches have been proposed to generate explainability maps directly from the raw-attention map corresponding to the cls token [23,24,25]. These approaches involve recording the self-attention maps generated by the self-attention heads of the last block in the ViT model during inference. The final explainability attention map can be obtained by averaging the attention vectors corresponding to the cls token in these self-attention maps. This explainability method is class-agnostic—similar to a saliency map—and is able to highlight several objects at the same time, even if they belong to different classes in the input.

However, the main challenge of these methods is the significant differences between the attention vectors of each head, which can introduce noise when taking the mean of the self-attention maps. Abnar et al. [26] proposed a method called attention rollout to solve the problem. They argued that in transformer-based models, the self-attention results need to be passed through a skip connection. Treating the raw-attention map as the sole source of explainable information would neglect the information processed during the skip connection [43].

Furthermore, relying solely on observing the raw-attention output of a single layer may not yield optimal results. Abnar et al. also proposed a linear combination of attentions to address this problem. Although this approach improves upon the noise problem associated with raw attention, it still faces challenges in accurately distinguishing between foreground and background regions.

Chefer et al. [27] proposed a novel explainability method that assigns relevance based on the deep Taylor decomposition principle. This method uses layer-wise relevance propagation (LRP) [44] to calculate the scores of each attention head related to the class token in each block. Combining the gradient information of the back-propagation gradient makes this method a class-specific explainability method. However, due to the existence of activation functions, gradients in the back-propagation process may suffer from issues such as gradient vanishing, resulting in sparse and noisy explainability maps as outputs.

## 3. Methods

This section provides an overview of the vision transformer and then introduces our proposed R-Cut method.

### 3.1. Vision Transformer (ViT)

The ViT model is a popular approach for image classification tasks that uses a transformer-based architecture. Given an input image X with resolution A×B. The network first splits X into several non-overlapping patches. If the size of each patch is p×p, the total number of patches will be $S=\frac{A\times B}{p\times p}$. Each patch is then flattened and linearly embedded into a token vector $t_{s}^{0}\in R^{1\times D},s\in \left[1,S\right]$, where *D* is the dimension of each token vector.

To enable the network to learn global features, a randomly initialized class token $t_{cls}^{0}\in R^{1\times D}$ is added to the tokens. Finally, the position embeddings are added to each of the tokens to form the input of the transformer block. If there are *L* cascaded transformer blocks, the input to each transformer block would be $t^{l}\in R^{(S+1)\times D}$, where l=1,⋯,L. In the vision transformer (ViT) architecture, each transformer block follows a specific arrangement of components. These components include layer normalization, an MHSA, a skip connection, and a multilayer perceptron layer (MLP). The input and output of each block consists of (S+1) discrete patch tokens; however, each attention head only processes subspace tokens t; if the number of heads in the MHSA is *H*, the dimension of t should be $D_{h}=D/H$ and $t\in R^{\left(S+1\right)\times D_{h}}$.

The MHSA of each layer $A_{h}^{l}$ is calculated as follows:(1)$$\begin{array}{cc} A_{h}^{l} & =\mathrm{Softmax}\left(\frac{f_{q}\left(t\right)f_{k}{\left(t\right)}^{T}}{\sqrt{d}}\right), \end{array}$$(2)$$\begin{array}{cc} & O_{h}^{l}=A_{h}^{l}\;\cdot \;f_{v}\left(t\right), \end{array}$$
where $f_{q}$, $f_{k}$, and $f_{v}$ are linear transformation layers in the *l*-th block. $A_{h}^{l}\in R^{\left(S+1)\times (S+1\right)}$ is the self-attention map of the input tokens from the *h*-th head in the *l*-th layer block. $O_{h}^{l}\in R^{\left(S+1\right)\times D_{h}}$ is the output of the head. The outputs $O_{h}^{l}$ of all heads are concatenated and fed into an MLP block.

From the last transformer block, the output class token $t_{cls}^{L}$ is used to obtain the category probability vector ViT(X); if there are *C* categories, $ViT\left(X\right)\in R^{1\times C}$.

The vector ViT(X) is generated as follows:(3)$$ViT\left(X\right)=\mathrm{Softmax}\left(\mathrm{MLP}\left(t_{cls}^{L}\right)\right),$$
where MLP denotes the classification head implemented by the MLP block. The corresponding class can be selected by taking the maximum value in the generated vector ViT(X).

### 3.2. Relationship Weighted Out and Cut

The method consists of two main stages, as depicted in Figure 2. In the first stage, called “Relationship Weighted Out”, the objective is to extract class-aware semantic information about the output results from the discrete intermediate tokens. The second stage, comprising fine-grained feature decomposition and named “Cut”, involves utilizing the class-specific intermediate vectors obtained in the first stage to construct a novel graph. Subsequently, graph cut operations are performed on the graph to derive foreground information that corresponds to the target. By leveraging these operations, the method generates a visual explainability map specific to the class based on the foreground information. The primary computational process is represented as follows:Generate alternative activation maps M from discrete tokens $t^{L}$;Generate perturbation maps P from alternative activation maps M and input image X;Calculate the class-aware weighting scores w based on perturbation maps P;Extract class-aware patch tokens $t^{c}$ based on the discrete tokens $t^{L}$ and class-aware weighting scores w;Construct a class-aware weighted graph G based on the class-aware patch tokens $t^{c}$;Get the class-aware solution eigenvector $y_{1}$ of the class-aware weighted graph G;Generate the explainability visualization map $L_{R-Cut}$ by partitioning the class-aware solution eigenvector $y_{1}$.

#### 3.2.1. Relationship Weighted Out

In this stage, we extract the class-aware semantic information related to the output results from the discrete patch tokens. Since directly extracting class-aware semantic information from the discrete tokens is challenging, we propose a perturbation-map-based approach to obtain the class-aware weight information. This approach consists of two main parts: generating alternative activation maps M and calculating the class-aware weighting scores w to extract class-aware patch tokens $t^{c}$.

Generate alternative activation maps M: As discussed in Section 3.1, ViT utilizes discrete tokens to convey information. The intermediate discrete tokens involved in the forward transmission process carry semantic information of the corresponding category as the network propagates category information during forward propagation. However, within each transformer block, there are multiple intermediate tokens. To address the interference caused by the skip connection, we select the output of the normalization layer after the skip connection in the last block to extract semantic information. We firstly generate the patch tokens $t_{S}^{L}$ by removing the last layer’s class token $t_{\mathrm{cls}}^{L}$ from the output of the last layer’s normalization $t^{L}\in R^{(S+1)\times D}$. Then, the alternative activation maps M will be generated from patch tokens $t_{S}^{L}$ as follows:(4)$$M=up(\frac{reshape\left(t_{S}^{L}\right)-\min\left(reshape\left(t_{S}^{L}\right)\right)}{\max\left(reshape\left(t_{S}^{L}\right)\right)-\min\left(reshape\left(t_{S}^{L}\right)\right)}),$$
where reshape(·) denotes a deserialization operation that can regroup the discrete patch tokens into a matrix map format, up(·) represents bi-linear interpolation for up-sampling with a scale factor of *p*, and $M\in R^{(A\times B)\times D}$.

Generate perturbation maps P: In this method, we consider M as *D* heat maps and perturb the original input image X through those heat maps to obtain perturbation maps $P\in R^{\left((A\times B\times 3)\times D\right)}$. The formula is shown as follows:(5)P=M⊙X,
where ⊙ denotes element-wise multiplication.

Calculate the class-aware weighting scores w: To compute the weight scores w for each perturbation map $P_{i}$, we input both the perturbation map matrix P and the original image X into the pre-trained ViT model. Then, we use the similarity between the output vectors to compute the weight scores w for each perturbation map $P_{i}$. A higher similarity between the output vectors indicates a stronger contribution of the corresponding perturbation map to the target class, which is calculated as follows:(6)$$w_{i}=\frac{\sum_{j=1}^{C}\left(ViT{\left(P_{i}\right)}_{j}\;\times \;ViT{\left(X\right)}_{j}\right)}{\sqrt{\sum_{j=1}^{C}ViT{\left(P_{i}\right)}_{j}^{2}}\;\times \;\sqrt{\sum_{j=1}^{C}ViT{\left(X\right)}_{j}^{2}}},$$
where w is a row vector of size *D*, *D* is the number of perturbation maps, ViT(·) denotes the output vector of the ViT model, and *C* represents the length of the output vector.

Extracte class-aware patch tokens $t^{c}$: Since the perturbation maps P are generated based on the original patch tokens $t_{S}^{L}$, the weight of each dimension of P regarding the original output result is equivalent to the weight of each dimension of the patch tokens $t_{S}^{L}$ regarding the original output result. Therefore, we can extract $t^{c}\in R^{S\times D}$ using the following formula:(7)$$t_{\mathrm{ij}}^{c}=w_{i}\times t_{\mathrm{Sij}}^{L}.$$

#### 3.2.2. Fine-Grained Feature Decomposition

In this section, we discuss how to finely partition the foreground and background information related to the category from the discrete tokens $t^{c}$ obtained from Section 3.2.1. In our previous research [29], we experimented with a simple method of summing all the dimensions of $t^{c}$ and reshaping the result to obtain the explainability feature map. The result shows that even when using such a simple method, we can also get a good result. However, this straightforward method does not consider the spatial position relationship of the discrete patch tokens, and it may not effectively address the issue of local discontinuities in the generated explainability map. To overcome these limitations and achieve more precise foreground–background partitioning, we propose a new method based on the graph cut technique discussed in Appendix B.

Firstly, we generate a class-aware weighted graph G=(V,e) using the class-aware patch tokens $t^{c}$. This graph considers both the direct relationship between nodes and the positional embedding relationship between the patch tokens. Next, we perform graph cut operations on this weighted graph to decompose it and obtain the corresponding class-specific eigenvector $y_{1}$. By leveraging the class-specific eigenvector $y_{1}$, we can identify the foreground vector $y_{1}^{c}$ associated with the target class.

Construct a class-aware weighted graph G: We generate the corresponding graph based on the class-aware patch tokens $t^{c}$. Specifically, we select the *S* class-aware patch token vectors ($t_{s}^{c}\in R^{1\times D},s=1,\cdots ,S$) in $t^{c}$ as the *S* nodes in the graph, resulting in V. Next, we define the edge $e_{ij}$ between two tokens $V_{i}$ and $V_{j}$ as the cosine similarity between them, incorporating both semantic and spatial information. By computing these similarities, we can obtain e. The formula for calculating the edge weights is as follows:(8)$$e_{ij}=\left\{\begin{array}{cc} \; & 1,\mathrm{if}\;\frac{\sum_{k=1}^{D}\left(V_{\mathrm{ik}}\;\times \;V_{\mathrm{jk}}\right)}{\sqrt{\sum_{k=1}^{D}{V_{\mathrm{ik}}}^{2}}\;\times \;\sqrt{\sum_{k=1}^{D}{V_{\mathrm{jk}}}^{2}}}\geq \varphi \\ \; & 0,else \end{array}\right.$$
where φ is a settable hyperparameter representing a constraint on the edges; we consider two nodes to be related only if the similarity between them exceeds φ.

Get the eigenvector $y_{1}$: To obtain the eigenvector $y_{1}$, we apply the normalized cut (Ncut) method described in Appendix B to partition the class-aware weighted graph G. This involves computing the generalized eigensystem (K−e)y=λKy of G and extracting the second-smallest eigenvector $y_{1}\in R^{1\times S}$. Appendix B provides a proof that the eigenvector $y_{1}$ is the Ncut of the class-aware solution of G, which is the class-aware vector we need corresponding to the target class.

Generate the explainability visualization map $L_{R-Cut}$ by partitioning the class-specific foreground and background information: To achieve this, we determine the splitting point by taking the mean value $\bar{y_{1}}=\frac{\sum_{i}^{S}y_{1}^{i}}{S}$ of the continuous eigenvector $y_{1}$. Then, we define the foreground set as $f=\{node_{i}|y_{1}^{i}\geq \bar{y_{1}}\}$ and the background set as $b=\{node_{i}|y_{1}^{i}<\bar{y_{1}}\}$.

To eliminate the interference brought by the background information, we set all nodes in the background set to 0. The class-specific vector $y_{1}^{c}$ is obtained by keeping the information of the foreground set unchanged.

Finally, we can obtain our class-specific explainability visualization map $L_{R-Cut}$ as follows:(9)$$L_{R-Cut}=\lambda *255*up\left(reshape\left(y_{1}^{c}\right)\right)+\left(1-\lambda \right)*X.$$
where λ represents the weight of the weighted-add, and ∗ represents multiplication.

## 4. Experiments

### 4.1. Experiment Setting

To verify the effectiveness of our class-specific post hoc visualization explainability method, we conducted three kinds of evaluation experiments (i.e., the point game [45], weakly supervised localization, and the perturbation test) with four SOTA explainability methods on ImageNet1K [28], i.e., raw-attention [23,24,25], rollout [26], grad-cam [31], and Hila’s method [27]. These methods belong to three different architectures: raw-attention and rollout are attention-based, grad-cam is gradient-based, and Hila’s method is a combination of attention and gradient-based approaches. We also performed three kinds of ablation experiments to verify the effectiveness of the different modules proposed in our methods. To further validate the applicability of our approach in real-world, complex scenarios, we also tested our method on the LRN dataset, which focuses on autonomous driving risk warning [29]. Lastly, we performed multiple sets of hyperparameter comparison experiments to ensure the rationality of the designed hyperparameters throughout our experiments.

#### 4.1.1. Datasets

We evaluated the proposed method (R-Cut) on the ImageNet1k [28] and LRN [29] datasets to verify the accuracy and effectiveness at generating explainability maps. Each of these two datasets brings different explainability map challenges. Figure 3 showcases samples from the ImageNet1K and LRN datasets.

ImageNet1k contains 1000 categories of image information, 1.28 million data points for training, and 50,000 datasets for variation. The 1000 object categories in ImageNet1k include common object classes found in daily life as well as relatively similar inter-class categories with small differences, such as numerous bird families and canines. This dataset contains many single-class but multi-object images in the validation set, which causes missed-detection problems for the generated explainability images. The biggest challenge for the fine-grained classes is the tendency of explainability maps to focus on discriminative regions due to the small inter-class differences. For example, in the case of birds like snowbirds and bulbuls, which differ mainly in the shape of their beaks, the explainability maps tend to cluster around the beak area.

The LRN dataset is a linguistic warning dataset we created for risk scenes in autonomous driving scenarios [29]. This dataset contains a total of 34,488 images and 10 linguistic cue categories. Each risk cue category consists of the type of risk object “car, cyclist, and pedestrian” and the general orientation information “ahead, ahead right, and ahead left” (e.g., “watch out for the pedestrian ahead right”). Therefore, even the same risk object in this dataset can be a different category depending on its location. The main challenges of this dataset are the complexity of the road scenarios and the influence of location information on the explainability maps.

#### 4.1.2. Implementation Details

In our experiments, we used the same pre-trained ViT base model as the backbone for our explainability map tests to ensure fairness. Given the ViT method’s previous success in image classification, we opted to maintain consistency with the hyperparameters used in ViT experiments. The chosen hyperparameters include: the input X is a three-channel 224×224 RGB image, each patch size of the patch embedding is 16×16, the number of heads in the MHSA layer is 12, and the number of transformer blocks is also 12. And we take 0.05 as the similarity threshold φ for constructing the graph. To ensure robust evaluation, we shuffle the dataset and then divide it into training, validation, and test sets at a 70:15:15 ratio for training and testing purposes. During our experiments, our method is numerically compared with previous SOTA methods. In subsequent tables, underlined numbers denote figures for comparison with our method. All our experiments are trained and tested on an RTX A6000 GPU with a batch size of 256 and 200 epochs of iterations during training.

### 4.2. Evaluation Matrices

For the quantitative experiments, we employed three commonly used evaluation metrics to assess the quality of explainability: the point game, IoU (intersection over union), and perturbation test.

#### 4.2.1. The Point Game Test

As described in [45], this method evaluates the correctness of the explainability map by checking whether the highest pixel value in the generated explainability image falls within the ground truth (GT) bounding box of the target object. If the highest pixel value is located within the GT bounding box, this indicates that the network’s explainability map correctly explains the object category.

The formula for this metric can be expressed as:(10)$$PG=\frac{1}{N}\sum_{i=1}^{N}\left[f\left(x_{i}\right)=y_{i}\right]\max j\in GT_{i}M_{ij}.$$
where *N* represents the total number of samples, $x_{i}$ refers to the input image of the *i*-th sample, $y_{i}$ denotes the ground truth label of the target category, *f* is the trained classification model, Mij represents the pixel value at position *j* in the generated explainability image, and $GT_{i}$ is the ground truth bounding box for the target category $y_{i}$.

The indicator function $\left[f\left(x_{i}\right)=y_{i}\right]$ is equal to 1 when the predicted label of the model *f* is the same as the true label $y_{i}$; otherwise, it is equal to 0. Therefore, this metric is a weighted average of classification accuracy and explainability, where the weight of explainability is determined by the highest pixel value Mij.

#### 4.2.2. The IoU Test

In the experiment on weakly supervised localization IoU conducted by [46], we followed a specific procedure. Firstly, the generated explainability feature map was up-sampled to match the size of the original image. Next, we set the threshold thres=0.2 to discard some background regions. Subsequently, the region within the explainability map was utilized to generate the predicted bounding box A by enclosing it with the minimum outer rectangle. Lastly, we employed intersection over union (IoU) as the evaluation metric to assess the quality of object-level localization achieved by the explainability feature map.

The formula for this metric can be expressed as:(11)$$IoU=\frac{A\cap B}{A\cup B}.$$
where B is the GT bounding box.

#### 4.2.3. The Perturbation Test

This test consists of two experiments: most relevant first perturbation (MRFP) and least relevant first perturbation (LRFP) as described by Hila’s method [27].

In MRFP, we begin by masking off the most relevant pixel part of the explainability map and generate the corresponding perturbation map. We then input the perturbation map into the trained model and observe the statistical change in the corresponding target’s confidence. A larger confidence change indicates better performance.

In LRFP, we preferentially mask off the most irrelevant part of the explainability map. We hope that the change in confidence is as small as possible because, in theory, the removed part does not belong to the target.

Throughout our experiments, we incrementally increase the proportion of masked pixels from 10% to 90%. We calculate the mean value of the confidence change as the actual confidence change value.

### 4.3. Results

#### 4.3.1. Performance on ImageNet1K

This section encompasses various types of qualitative and quantitative analysis on the ImageNet1K dataset. For our qualitative analysis, we conducted post hoc explainability visualization experiments on single-class single-object images, single-class multi-object images, multi-class single-object images, and multi-class multi-object images. Regarding our quantitative analysis, we employed three different tests: the point game, IoU, and perturbation test.

Figure 4 presents the performance of our R-Cut method and other methods on the Imagenet1k dataset for single-class single-object images, single-class multi-object images, and fine-grained images (the bird family) with small inter-class differences. The explainability visualization experiments were conducted separately for regular-shaped objects and irregularly shaped objects in order to ensure fairness.

As shown in Figure 4, the raw-attention and rollout methods exhibit more background noise, while the grad-cam method accurately locates the object but only highlights the discriminative regions. Hila’s method is relatively effective at activating the corresponding regions but still exhibits local discontinuities in the explainability map. In contrast, our R-Cut method eliminates the background noise and mitigates the discriminative region problem in fine-grained categories (d) and (e). Moreover, our method accurately identifies all objects in single-class multi-object images (c) and (f). To demonstrate that our method is a class-specific approach, we conducted comparative explainability visualization analysis on multi-classes images, such as the classic “dog and cat” and “elephant and zebra”. The purpose is to show different corresponding explainability visualizations for different object categories within the same image.

As shown in Figure 5, the raw-attention method and rollout method are class-agnostic methods, while the grad-cam method and Hila’s method can visualize different classes of objects but suffer from background noise interference and local discontinuity problems. In contrast, our method not only can visualize the explainability maps of different classes but can also generate regions of explainability maps that can effectively mask objects. Our R-Cut method can also visualize and explain multi-class multi-object images clearly.

Point game test results: Table 1 shows the results of the point game localization experiments on the ImageNet1k dataset with explainability maps. It is evident that our method outperforms the SOTA method by 2.36% on the ImageNet1K dataset when utilizing GT categories. Additionally, without the knowledge of GT categories, our method still achieves a notable improvement of 1.61% compared to the previous SOTA method. These results emphasize the effectiveness and superiority of our method for accurately localizing objects within the ImageNet1K dataset.

IoU test results: Table 2 presents the results of the pixel-level explainability localization IoU experiments. Our method demonstrates a significant improvement of 4.5% (with GT) and 4.09% (without GT) on the ImageNet1K dataset when compared to the previous method by Hila. These results validate the enhanced completeness and explainability of our method for localizing object pixels.

Perturbation test results: The above two test metrics are artificially defined metrics; in order to get a good explanation to reflect the actual regions that the model is using, we also conducted a perturbation test. As showed in Table 3. For MRFP, wherein we mask off the most relevant region related to the model’s prediction, we expect a high confidence change in the model’s prediction about the corresponding category. Our method demonstrates a significant improvement of 3.6% compared to Hila’s SOTA method. For LRFP, we believe that the masked-out region should be irrelevant to the model’s prediction, so we hope that the impact on confidence is as small as possible. We can see that our method‘s LRFP result is 15.69%, which is 1.22% lower than Hila’s method.

Both qualitative and quantitative results show that our explainability visualization method is much better than the previous SOTA method on the ImageNet1K dataset.

#### 4.3.2. Performance on LRN Dataset

To verify the effectiveness of our method in complex scenarios, we also performed qualitative and quantitative analyses on the hazard warning dataset LRN [29] for autonomous driving scenarios. Figure 6 shows the explainability visualization results of our R-Cut method and other methods on the LRN dataset. We visually post hoc explained each of the three risk categories: dangerous vehicle, dangerous cyclist, and dangerous pedestrian. The visualizations clearly demonstrate that our method can visually explain the situation accurately even in traffic scenes with complex backgrounds.

Point game test results: Table 4 shows the results of our method and other SOTA methods in point game localization experiments on the LRN dataset with the generated explainability maps. Our method outperforms the previous SOTA method with significant improvements. Specifically, our method achieves a remarkable improvement of 21.44% without GT and 21.67% with GT compared to the previous SOTA method. These results demonstrate the superior object-level explainability localization performance of our method in driving scenes.

IoU test results: Table 5 shows the results of the pixel-level explainable localization IoU experiments. Our method and other baselines were evaluated on the LRN dataset. It is observed that our method achieved a notable improvement of 5.34% without the GT category and 5.56% with the GT category compared to Hila’s method. These results demonstrate that our method can more completely explain the pixels that belong to the risk object.

Perturbation test results: In the MRFP test, we aimed to observe the impact on the output perturbation map confidence after the perturbation, and we expected to see a significant impact. As shown in Table 6, our method outperformed Hila’s method by 5.73% in this test. In the LRFP test, our method outperformed Hila’s method with a reduction of 1.62%.

#### 4.3.3. Ablation Test

To validate the efficacy of our two proposed modules, we conducted qualitative and quantitative experiments to evaluate three method variants: (1) only “Relationship weighted out”, (2) only “Cut”, and (3) R-Cut. As shown in Figure 7, the “Relationship weighted out” method includes a class-aware function, but it does not consider spatial location relationships, which leads to local discontinuities. For example, the chest position of the dog is not activated in the R-Out column in Figure 7a. On the other hand, the Cut method generates locally dense explainability maps by considering location, texture, and color information during the graph decomposition process, but it remains a class-agnostic map. Moreover, since color information is considered in the computation process, the Cut method considers the brown desktop and the black drawer in Figure 7b as not belonging to the same entity. In contrast, the R-Cut method can generate both class-aware and dense explainability maps.

Table 7 shows the performance of the three method variants on the point game, IoU, and perturbation test experiments, and it is evident that the R-Cut method achieves the best results. The experimental results demonstrate that only R-Cut can generate a fine-grained class-specific explainability map.

Furthermore, we present the localization results of our method for the point game test with different hyperparameter φ values to demonstrate the rationality of our chosen values. As depicted in Table 8, it is evident that our method achieves the best performance when φ=0.05.

## 5. Discussion

Based on multiple previous experiments, it is evident that our method stands out compared to others. Not only does it generate class-specific explainability maps tailored to multi-object categories, but it also yields more refined results. The heatmaps produced are clearer and more continuous and do not have the occurrence of solely detecting discriminative regions in fine-grained images. Clearly, our approach provides effective and rational explainability for the model. While our algorithm demonstrates remarkable explainability results on both the ImageNet and LRN datasets, our study also reveals certain limitations. Primarily, our method necessitates substantial computational overhead, which is compounded by its intricate procedural steps. As a consequence, each explainability iteration demands a significant time investment. Hence, our forthcoming endeavors are focused on optimizing the algorithm’s speed to alleviate these concerns. Furthermore, we recognize that our current explainability framework overlooks applications within the multimodal domain. As our next trajectory, we aim to delve deeper into the realm of multimodal explainability with the aim of more nuanced explorations and implementations in this domain.

## 6. Conclusions

This paper introduces a novel post hoc visualization explainability method for transformer-based image classification tasks. Our method addresses the crucial need for trust and understanding in classification results. Through our proposed “Relationship weighted out” module, we can obtain class-specific information from intermediate layers, enhancing the class-aware explainability of the discrete tokens. Additionally, our “Cut” module enables fine-grained feature decomposition. By combining the two modules, we can generate dense class-specific visual explainability maps.

We extensively evaluated our explainability method on the ImageNet dataset, conducting both qualitative and quantitative analyses. Furthermore, we tested the explainability of our method in complex backgrounds by performing numerous experiments on the LRN dataset for automatic driving danger alerts.

The results of both sets of explainability experiments demonstrate significant improvement of our method compared to previous SOTA approaches. Additionally, through ablation explainability experiments, we provide further validation of the effectiveness of the different modules proposed in our method.

Overall, our method not only enhances trust in transformer-based image classification but also contributes to the comprehension of the model, benefiting downstream tasks. In the future, we plan to extend our work to perform explainability experiments on multi-modal tasks.

## Figures and Tables

**Figure 1 sensors-24-02695-f001:**
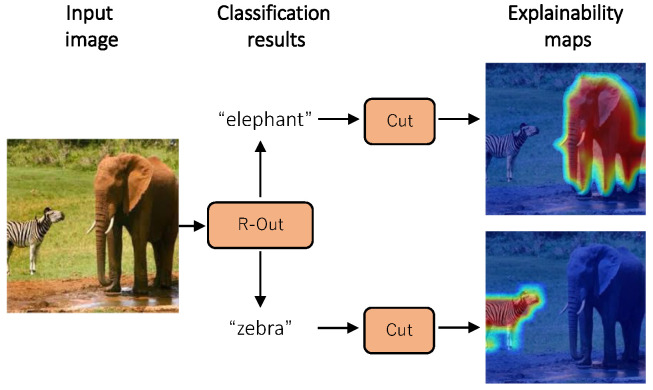
Overview of our method. Our method can generate a class-specific post hoc explainability map for different results after the “R-Out” and “Cut” steps.

**Figure 2 sensors-24-02695-f002:**
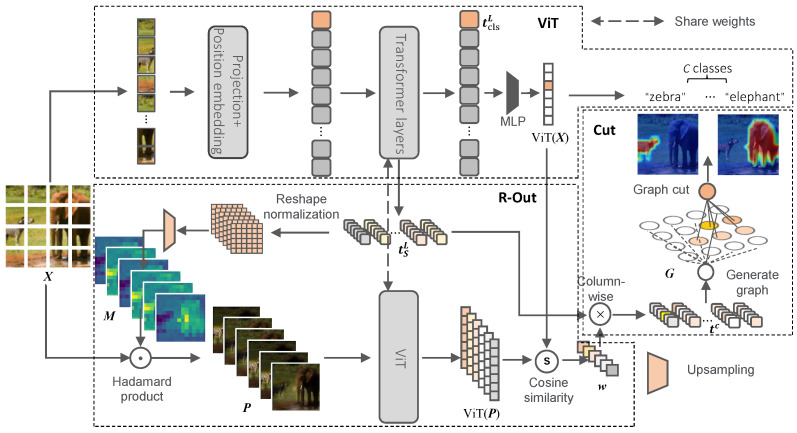
Overall architecture for our method. First, we extract $t_{S}^{L}$ from ViT. Next, we use our “R-Out” module to extract class-aware token $t^{c}$. We then employ the “Cut” module for fine-grained feature decomposition. By combining these modules, we obtain class-specific explainability maps.

**Figure 3 sensors-24-02695-f003:**
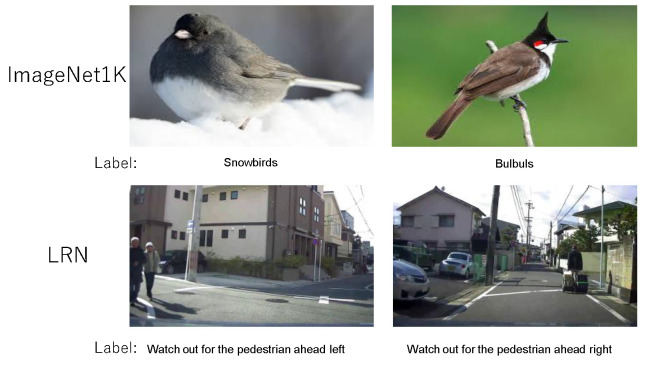
Dataset examples. Figure displays two closely related bird species from the ImageNet1K dataset and two closely related categories of hazardous pedestrians from the LRN dataset.

**Figure 4 sensors-24-02695-f004:**
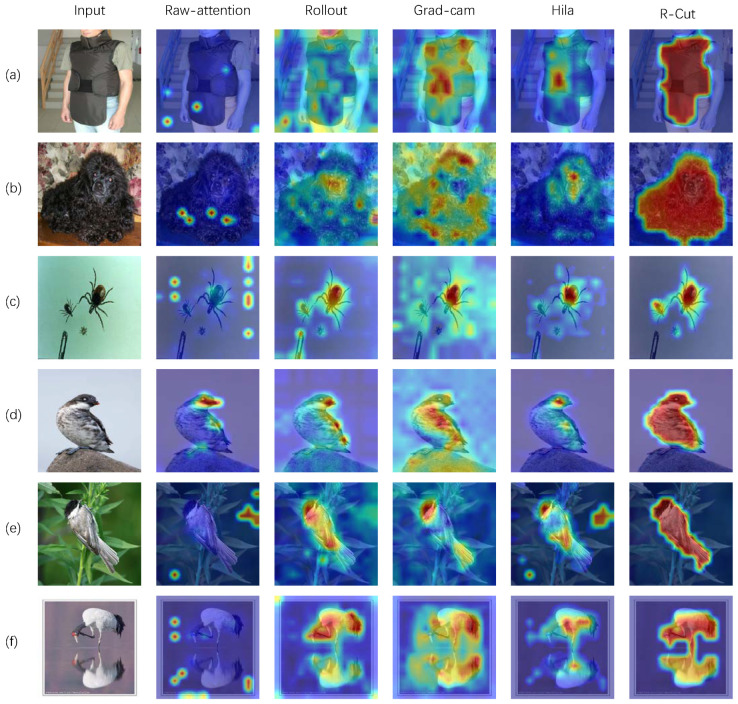
Single-class explainability visualization test for ImageNet1k: (**a**–**c**) represent the normal categories, (**d**–**f**) represent the fine-grained categories, and (**c**,**f**) represent the explainability visualization results for single-class multi-object images.

**Figure 5 sensors-24-02695-f005:**
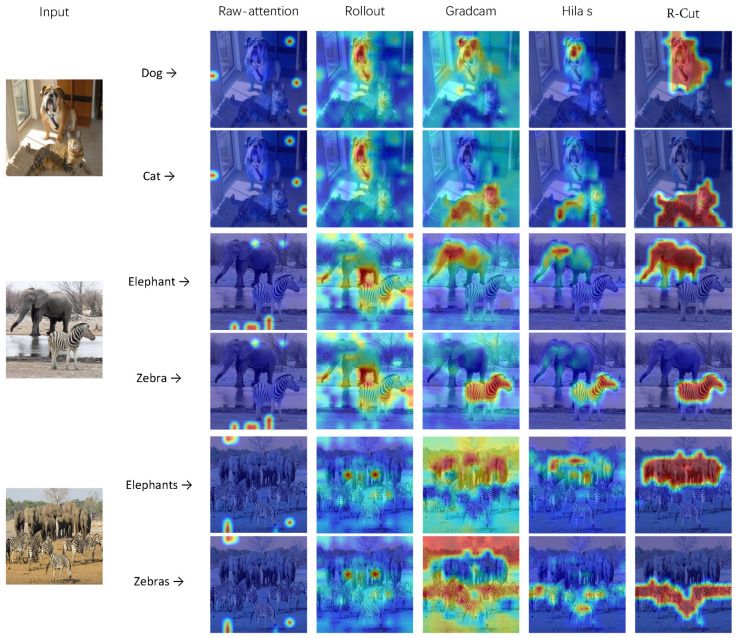
Multi-class explainability visualization test on ImageNet1K: “Dog and Cat” and “Elephant and Zebra” represent the multi-class single-object explainability visualization results; “Elephants and Zebras” represents the multi-class multi-object explainability visualization results.

**Figure 6 sensors-24-02695-f006:**
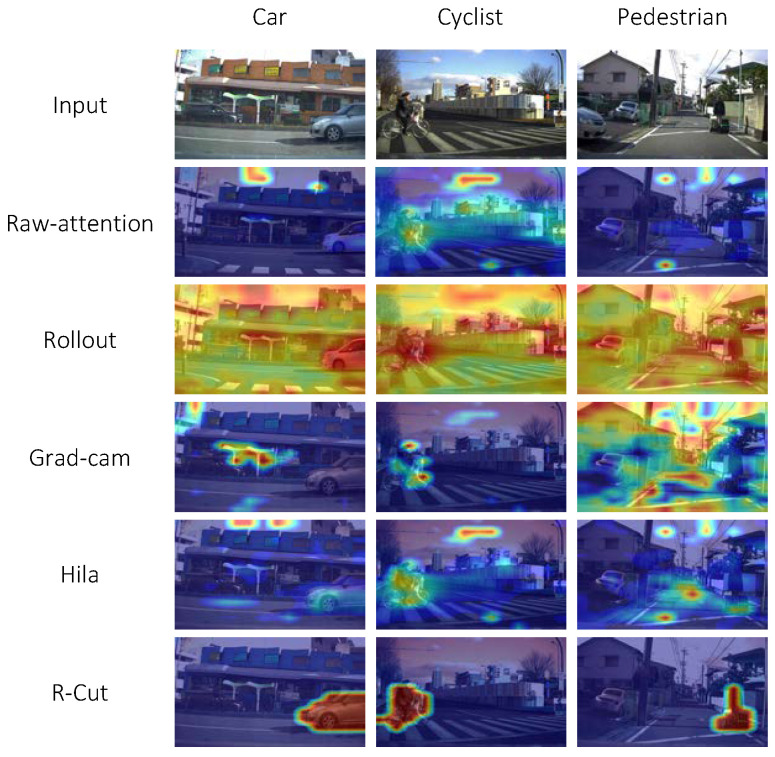
Explainability visualization results for the LRN dataset. In this result, “car” represents the warning “Watch out for the car ahead right”; “cyclist” represents the warning, “Watch out for the cyclist ahead left”; “pedestrian” represent the warning “Watch out for the pedestrian ahead right”.

**Figure 7 sensors-24-02695-f007:**
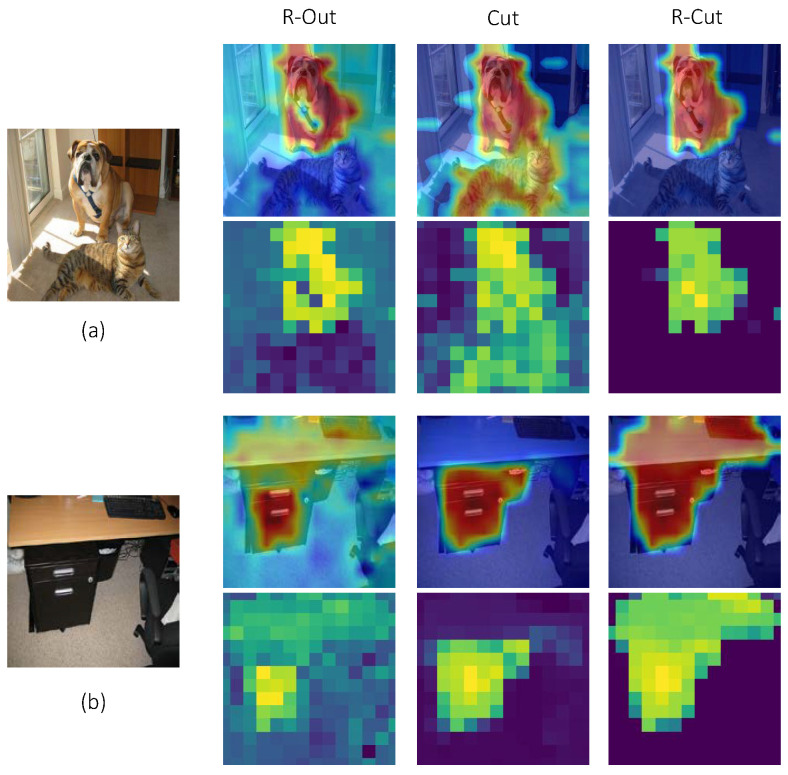
Ablation tests of three method variants. (**a**) needs to demonstrate explainability for the region of the dog, while in (**b**), interpretability needs to be shown for the entire table area. Plots in even rows represent the heatmaps of the corresponding explainability maps.

**Table 1 sensors-24-02695-t001:** Point game test on ImageNet1K dataset.

	ImageNet1k	
	Pre	GT
Raw-attention	59.21	59.21
Rollout	70.33	70.33
Gradcam	71.70	74.05
Hila	75.50	77.73
R-Cut	77.11 (↑1.61)	80.09 (↑2.36)

**Table 2 sensors-24-02695-t002:** Weak object detection IoU on ImageNet1K.

	ImageNet1k	
	Pre	GT
Raw-attention	46.37	46.37
Rollout	52.91	52.91
Gradcam	51.95	53.14
Hila	53.41	54.29
R-Cut	57.50 (↑4.09)	58.79 (↑4.50)

**Table 3 sensors-24-02695-t003:** MRFP and LRFP tests on ImageNet1K.

	ImageNet1k	
	MRFP	LRFP
Raw-attention	45.57	24.36
Rollout	53.31	21.01
Gradcam	52.23	26.42
Hila	53.47	16.91
R-Cut	56.91 (↑3.44)	15.69 (↓1.22)

**Table 4 sensors-24-02695-t004:** Point game test on LRN dataset.

	LRN	
	Pre	GT
Raw-attention	33.56	33.56
Rollout	41.78	41.78
Gradcam	51.56	53.22
Hila	50.22	52.33
R-Cut	73.00 (↑21.44)	74.89 (↑21.67)

**Table 5 sensors-24-02695-t005:** IoU test on LRN dataset.

	LRN	
	Pre	GT
Raw-attention	24.11	24.11
Rollout	32.55	32.55
Gradcam	44.75	46.67
Hila	45.56	47.00
R-Cut	50.90 (↑5.34)	52.56 (↑5.56)

**Table 6 sensors-24-02695-t006:** MRFP and LRFP tests on LRN dataset.

	LRN	
	MRFP	LRFP
Raw-attention	33.16	31.3
Rollout	37.92	35.42
Gradcam	42.53	29.71
Hila	44.39	20.38
R-Cut	50.12 (↑5.73)	18.76 (↓1.62)

**Table 7 sensors-24-02695-t007:** Ablation tests of three method variants.

	**Point Game Test**			
	**R-Out**		**Cut**	**R-Cut**
ImageNet1K	78.15		77.11	**80.09**
LRN	74.22		73.88	**74.89**
	**IoU Test**			
	**R-Out**		**Cut**	**R-Cut**
ImageNet1K	55.27		52.46	**58.79**
LRN	49.33		35.33	**52.67**
	**Perturbation Test**			
		**R-Out**	**Cut**	**R-Cut**
ImageNet1K	MRFP	54.44	54.37	**56.91**
	LRFP	17.72	19.86	**15.69**
LRN	MRFP	48.53	47.82	**50.12**
	LRFP	19.92	21.4	**18.77**

**Table 8 sensors-24-02695-t008:** Performance of point game test with different hyperparameter φ values.

	0	**0.05**	0.1	0.15	0.2	0.25
ImageNet1K	79.33	**80.09**	78.29	77.92	77.24	76.75

## Data Availability

No new data were generated for our research; instead, we utilized two public datasets at https://www.image-net.org/ (accessed on 19 January 2022) to conduct our experiments.

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
