# Peer review of "R-Cut: Enhancing Explainability in Vision Transformers with Relationship Weighted Out and Cut"

_sensors, 2024, doi:10.3390/s24092695_

Round 1
Reviewer 1 Report
Comments and Suggestions for Authors
R-Cut: Enhancing Explainability in Vision Transformers with
Relationship Weighted Out and Cut
The contribution of this article is very interesting, but I have the following observations.
-
In line 204 and line 209 and 216. I correct to use the notation reshape(·), up(·) and ViT(·)? In necessary the (·).
-
Line 209. Review the expression p. P. It is confused.
-
In Eq. 5. What mathematical operation does the operator ⊙ represent? Add detail.
-
The authors must add pseudocode or the steps to follow for the implementation.
-
In Eq. 9. What mathematical operation does the operator (*) represents? Add detail.
-
Section 4.1.1. Datasets. I suggest adding some illustrations for the two data sets used in the experimentation.
-
Section 4.1.2. Implementation Details. It is important to mention the number of experiments carried out for the proposal and for the methods that are compared. Additionally, the author should add the % of data used for training and testing.
-
Section 4.1.2. Implementation Details. Explain how the given hyperparameters were determined. Why are these used
-
The results presented in all tables for R-Cut are confusing. The authors mention that it is better for R-Cut but the values ​​presented are the lowest.
-
In the article, the authors use IoU and mIoU. Are these correct?
-
According to the numerical results presented in the Tables. It is not clear how the percentage improvement is calculated. Please add an explanation about this.
-
The main contribution of this article is to improve explainability. What special features of this approach make the method more explainable? The authors should add some notes in the discussion.

Author Response
We are deeply grateful for your thorough review and the invaluable feedback and suggestions you have provided. Your feedback has truly enhanced the quality of our paper, and we are sincerely grateful for your comments. In light of your insightful comments, we have carefully revised our manuscript as follows:
"Please see the attachment."

Reviewer 2 Report
Comments and Suggestions for Authors
This paper presents a method to enhance the explainability of Transformer-based image classification models. They introduce two modules: the “Relationship Weighted Out" and the “Cut" modules. The “Relationship Weighted Out" module focuses on extracting class-specific information from intermediate layers, enabling us to highlight relevant features. The “Cut" module performs fine-grained feature decomposition, taking into account factors such as position, texture, and color. By integrating these modules, generate dense class-specific visual explainability maps. The method proposed in this paper has good theoretical innovation and practical engineering application value. Additionally, there are some questions and areas requiring revision in the manuscript. The main concerns about this manuscript are as follows:
1. The highlight of this paper lies in "interpretability experiments", and the author should further highlight this aspect in their contribution to this paper.
2. The author mentions on page 5 of the article: where MLP notes the classification head implemented by the MLP block, But there is no marking of the MLP block in Figure 2.
3. What operator is ⊙ in formula (5)? The author should indicate it in the article.
4. The M in formula (4) is essentially a normalization operator, while the up (M) in formula (5) is an up-sampling operator. The author should explain clearly how to up-sampling?
5. The discussion section of the article looks more like the author's future work. Suggest rewriting this section.
Author Response

(The authors gave the same response as above.)

Reviewer 3 Report
Comments and Suggestions for Authors
This manuscript proposes a novel method for enhancing the explainability in vision transformers for image classification. The method consists of two major modules, namely "relationship weight out" and "cut", to determine class-specific information. Enhancing the explainability of DL-based image classification methods is an important topic, especially for transformers which are relatively new. Overall, the manuscript is well written. A thorough literature review on the previous explainability efforts on DL networks is provided together with limitations of the existing methods and motivation of the current work. The proposed method is described clearly with sufficient details. The design of the experiments is appropriate, and the results are presented with proper graphs and data. The limitation of the method is also mentioned, and the conclusion is supported by the results.
There is only a minor issue I found from the manuscript: Tables 1-6 seem to have incomplete digits shown due to insufficient table width. Please correct.
Author Response
We are deeply grateful for your thorough review and the invaluable feedback and suggestions you have provided. We are heartened by your positive reception of our research, and we are more than willing to consider the adjustments you have proposed. In light of your insightful comments, we have carefully revised our manuscript as follows:
concern1:There is only a minor issue I found from the manuscript: Tables 1-6 seem to have incomplete digits shown due to insufficient table width. Please correct.
response:We are deeply grateful for your thorough review and the invaluable feedback and suggestions you have provided. We truly appreciate your valuable feedback. As you pointed out, the inappropriate table width led to the loss of some data. We've addressed this issue by fixing the table width in tables 1-6 of our latest version. Thank you very much sincerely.
Round 2
Reviewer 1 Report
Comments and Suggestions for Authors
All previous revision has been completed and the new version of the paper has been improved.
I only have one suggestion: on line 356, the author should add the equation number for IoU.
Author Response
Thank you very much for offering invaluable insights into our article. We've noted that the IOU formula lacked numbering, and thus, in the latest version of our paper, we've included the appropriate numbering for it. Your feedback has been immensely valuable in enhancing the caliber of our paper, and we sincerely appreciate it.